# Correlation Between Instrumented Timed Up and Go Test Performance and Muscle Strength in Knee and Hip Osteoarthritis: An Exploratory, Cross-Sectional Study

**DOI:** 10.3390/s26010142

**Published:** 2025-12-25

**Authors:** Mathilde Pelletier Visa, Lech Dobija, Anargyros Verdilos, Aurélien Mullez, Vivien Reynaud, Paul Gignoux, Frederic Costes, Emmanuel Coudeyre

**Affiliations:** 1Service de Médecine Physique et de Réadaptation, Centre Hospitalier Universitaire de Clermont-Ferrand, 63000 Clermont Ferrand, France; 2Unité de Nutrition Humaine, INRAE, Université Clermont Auvergne, 63000 Clermont-Ferrand, France; 3Plateforme Mobilité, Centre Hospitalier Universitaire de Clermont-Ferrand, 63000 Clermont Ferrand, France; averdilos@chu-clermontferrand.fr; 4Direction de le Recherche Clinique et de l’Innovation, Centre Hospitalier Universitaire de Clermont-Ferrand, 63000 Clermont Ferrand, France; 5Service de Médecine Physique et de Réadaptation, Centre Médical Germaine Revel, 69440 Chabanière, France; 6Service de Médecine du Sport et des Explorations Fonctionnelles, Centre Hospitalier Universitaire de Clermont-Ferrand, 63000 Clermont Ferrand, France

**Keywords:** chronic disease, hip osteoarthritis, knee osteoarthritis, muscle strength, timed up and go test

## Abstract

**Highlights:**

**What are the main findings?**
iTUG parameters demonstrated weak-to-moderate correlations with quadriceps and hamstring strength in knee osteoarthritis, and moderate-to-strong correlations in hip osteoarthritis.Vertical push-off power, walk duration, and total iTUG duration showed the strongest associations in hip osteoarthritis population.

**What are the implications of the main findings?**
The iTUG test could provide a simple and accessible estimation of lower-limb muscle strength when isokinetic evaluation is not available.The classic stopwatch-based TUG may offer comparable clinical insights to iTUG-derived metrics, supporting its use in daily rehabilitation settings.

**Abstract:**

Muscle weakness is a key contributor to functional limitation in individuals with hip (HOA) or knee osteoarthritis (KOA). Identifying accessible and objective tools to estimate muscle strength could improve clinical assessment and rehabilitation monitoring. This exploratory cross-sectional study investigated the relationship between parameters derived from the instrumented Timed Up and Go test (iTUG), and isokinetic quadriceps and hamstring strength in individuals with KOA or HOA. Thirty-six participants with KOA and 21 with HOA were assessed. Quadriceps and hamstring peak torques were measured at 60°/s using an isokinetic dynamometer. The iTUG was performed with a sacrum-mounted inertial measurement unit. Correlations between iTUG parameters and lower-limb muscle strength were weak-to-moderate in KOA (r = −0.398 to −0.516, *p* < 0.05) and moderate-to-strong in HOA (r= −0.537 to −0.843, *p* < 0.05). Total iTUG duration strongly correlated with the traditional stopwatch TUG (r = 0.91, *p* < 0.01), suggesting that the classic stopwatch-based TUG may offer comparable clinical insights to certain iTUG-derived temporal metrics. The iTUG test reflects lower-limb muscle weakness, especially in HOA, with vertical push-off power and temporal metrics showing the strongest relationships. These results support the use of iTUG as an accessible tool for estimating muscle weakness when isokinetic testing is unavailable. However, these findings should be interpreted with caution given the small sample size.

## 1. Introduction

Musculoskeletal disorders are the leading cause of disability worldwide, and, according to a recent Global Burden of Disease study, 1.71 billion people worldwide live with a musculoskeletal disorder [1]. Knee (KOA) and hip osteoarthritis (HOA) are highly prevalent musculoskeletal disorders associated with pain and disability that increase morbidity [2]. Quadriceps, hamstring, and hip muscle strength are considerably impaired in people with KOA [3]. Hip osteoarthritis also reduces the strength of both knee and hip muscles, and may impact the contralateral limb [4]. Muscle strength is a component of physical capacity that contributes to balance and functional performance, including stair negotiation, walking, and transfers. The gold standard method for muscle strength evaluation is isokinetic dynamometry [5]. This technique provides a reliable, open-chain evaluation of muscle strength, controlling for movement speed, range of motion, and contraction type. Optoelectronic systems combined with force platforms can provide muscle force moments during functional activities [6], providing relevant information regarding muscle strength utilization during activities of daily living. However, both isokinetic and optoelectronic systems have limited accessibility in clinical settings due to their high costs and time-consuming procedures. The Timed Up and Go test (TUG) is a simple tool, widely used in clinical settings for functional evaluations. The time taken to stand up from a chair, walk 3 m, turn around, walk back to the chair, and sit down is recorded in seconds using a stopwatch [7]. The TUG test is reliable [8,9], but does not provide detailed information on the potential mechanisms underlying poor outcomes. Such information would be valuable for guiding clinical management.

In recent years, inertial measurement units (IMUs) have been increasingly used to assess posture and gait [10,11]. These portable systems are lightweight and compact, incorporating three-dimensional accelerometers and gyroscopes [12]. IMUs are being increasingly used to provide additional measurements during functional tests commonly used in clinical practice. The TUG test has been combined with IMUs [12,13,14] and is referred to as the instrumented TUG (iTUG). IMUs appear to be a practical option for assessing functional mobility, due to their increasing availability, especially for clinical professionals such as physiotherapists and nurses, who may not have a strong technical background. Several studies have evaluated the use of inertial measurement units with various sensor placements in older adults as well as in individuals with KOA and HOA [13,15,16]. However, no studies have examined the correlation between isokinetic quadriceps and hamstring muscle strength and the parameters derived from the mTUG^®^ inertial measurement unit in individuals with KOA and HOA. Such an analysis could provide insight into the clinical relevance of using this device to detect muscle weakness.

Therefore, this study evaluate the ability of the mTUG^®^ inertial measurement unit to detect muscle weakness in individuals with KOA and HOA. The primary objective was to evaluate the correlation between vertical acceleration (Root Mean Square of the vertical acceleration during sit-to-walk) during the iTUG and isokinetic quadriceps strength at 60°/s. Vertical acceleration during the sit-to-walk phase was selected as the primary parameter because it reflects the ability to generate vertical thrust forces when rising from a chair, a function closely dependent on quadriceps and hamstring strength. Previous studies have reported that individuals with lower muscle strength exhibit reduced vertical acceleration amplitudes during sit-to-stand tasks, supporting its use as a surrogate marker of thigh muscle strength [17,18]. Secondary objectives were to explore associations between additional iTUG parameters (total iTUG duration, vertical angular velocity range during sit-to-walk, vertical push-off power during sit-to-walk, and walk duration) and isokinetic quadriceps and hamstring muscle strength.

## 2. Methods

### 2.1. Study Design

A single-center, cross-sectional, exploratory study, involving non-invasive functional assessments in people with KOA and HOA was conducted in the Clermont-Ferrand University Hospital.

### 2.2. Ethical and Regulatory Aspects

The study was conducted in accordance with the Declaration of Helsinki, as revised in 1983, and followed the ethical principles of the World Medical Association (www.wma.net, accessed on 1 December 2019). Ethical approval was obtained from an ethics committee (Comité de Protection des Personnes), North-West III, Reference number: 20.02.13.58801) on 3 March 2020. All participants received oral and written information about the study protocol. Written informed consent was obtained before their inclusion. The investigator addressed all participants’ questions regarding the study protocol before inclusion. Participants were informed that they could withdraw from their participation at any time at their own discretion. All personal data collected during the study were anonymized and treated confidentially. Only authorized study personnel had access to the data, and all necessary measures were taken to ensure participant confidentiality throughout the study and in the dissemination of results.

The study was registered on ClinicalTrials.gov: NCT04368858. The study report adheres to the Strengthening the Reporting of Observational Studies in Epidemiology (STROBE) guidelines [19].

### 2.3. Setting

The study was conducted at the Mobility Exploration Unit between August 2020 and November 2021. People were recruited from the Mobility Exploration Unit and the Department of Sports Medicine and Physical Medicine and Rehabilitation at the Clermont-Ferrand University Hospital.

### 2.4. Participants

The inclusion criteria were age between 18 and 90, impaired mobility due to advanced KOA or HOA (surgical or medical stage), affiliation with the French social security system, and the ability to provide written informed consent. People with progressive psychiatric illness (e.g., active psychosis) or other serious illness (e.g., decompensated heart failure, progressive cancer in terminal phase), poorly controlled or unstable cardiovascular pathology, osteoarticular pathology that could significantly impair functional performance, people under guardianship, curatorship, deprivation of liberty or safeguard of justice, and pregnant women were not included.

### 2.5. Procedures

All participants underwent the iTUG test using the mTUG^®^ inertial unit (mHealth Technologies, Bologna, Italy), along with an isokinetic muscle strength assessment. All tests were conducted by the same two investigators (AV, VR) on the same day and in the same standardized order. The same testing procedure was applied to all participants, regardless of whether they had KOA or HOA. The participants performed a 5 min warm-up on a cycle ergometer at 30 watts before the evaluations. A 15 min rest period was provided between the iTUG and muscle strength measurements.

#### 2.5.1. Muscle Strength Measurement

Quadriceps and hamstring muscle strength were assessed bilaterally using a Humac Norm Cybex 6000™ isokinetic dynamometer CSMi, Lawrence, MA, USA, in concentric mode at a velocity of 60°/s, over a 0–90° range of motion, adjusted if joint limitations were present. The evaluation procedure followed standard recommendations [20,21]. Before each measurement, a gravity compensation procedure was performed. Participants first completed three submaximal warm-up repetitions to familiarize themselves with the procedure, followed by three maximal voluntary concentric quadriceps and hamstring contractions. Verbal encouragement was provided during each effort to optimize performance. A 30 s rest period was given between each trial. The best result was taken and normalized to the person’s body mass. Both limbs were tested in all participants. The affected limb was identified based on clinical findings (pain, stiffness, functional limitation) and confirmed by radiographic evidence of osteoarthritis.

#### 2.5.2. Instrumented TUG

The iTUG was performed in accordance with the recommendations of the American Thoracic Society and recorded on a data sheet [22]. The test took place in a spacious examination room. A chair with armrests was positioned 3 m away from a floor marker indicating a U-turn. Participants were instructed to initiate the test upon hearing an auditory signal: they stood up from the chair, walked at a comfortable pace to the marker, turned, walked back, and sat down again. The test was considered complete once the person’s back touched the back of the chair. The test was explained verbally by the investigator; however, no practice trial was performed. No instructions were given concerning the use of the arms for the rising or sitting phases, as recommended. The same 45 cm chair with armrests was used for all participants. The test and device are shown schematically in Figure 1. A traditional stopwatch TUG measurement was performed simultaneously by the investigator.

The mTUG^®^ system is a medical device based on IMUs, designed to analyze kinematics during the TUG test [23]. It includes a single inertial sensor placed on the upper part of the sacrum attached with an adjustable elastic belt. The sensor is connected via Bluetooth to a smartphone (Samsung Galaxy S4, Samsung Group Seoul, Seoul, Republic of Korea, Android 5.0.1), which is the control and processing unit, held by the investigator. The inertial unit comprises a triaxial accelerometer, a triaxial gyroscope, and a triaxial magnetometer. The sampling frequency is 200 Hz. The sensor weight is negligible (22 g).

During the iTUG, the IMUs capture the duration of each TUG phase automatically (sit-to-walk, walking, half-turning, and stand-to-sit), along with the corresponding 3D velocities and accelerations. The processes of filtering, segmenting the movement into phases, and defining the measurement parameters were based on previous studies [24,25].

### 2.6. Data Sources and Measurement

To describe the clinical characteristics of the participants, data were collected on age, sex, type of residence, type of pathology, body mass, height, and Body Mass Index (BMI). Functional and symptom-related data were collected using standardized, validated questionnaires:-Western Ontario and McMaster Universities Osteoarthritis Index (WOMAC): Questionnaire used for KOA. This 24-item questionnaire calculates three subscale scores for pain, stiffness and physical function. Participants rate their difficulty with each item on a Likert scale. The total score ranges from 0 (no symptoms) to 96 (major disability in daily life) [26].-Knee Injury and Osteoarthritis Outcome Score (KOOS): administered to participants with KOA only. This 42 items questionnaire, which can be grouped into 5 sub-categories assessing symptoms, pain, function, activity and quality of life. The standardized score ranges from 0 to 100. A score of 100 reflects the best possible outcome, and 0 the worst [27].-Hip Disability and Osteoarthritis Outcome Score (HOOS): The HOOS was completed by patients with HOA. It comprises 40 items, with 5 sub-scores for pain, symptoms, daily activities, sporting activities and quality of life. Responses are rated on a Likert scale. A standardized score ranging from 0 (extreme problems) to 100 (no problems) is calculated separately for each sub-score.

Data collected from the iTUG evaluation were obtained from raw tri-axial accelerometer and gyroscope signals, which were processed through validated algorithms that automatically segment the iTUG into sit-to-walk, walking, turning, and walk-to-sit phases [24]. These algorithms identify movement onset, transitions between phases, and movement termination, enabling the extraction of time- and amplitude-based metrics. Following iTUG parameters were recorded:-Total iTUG duration [s], which represents the time interval between the algorithm-detected onset of the sit-to-stand phase and the return to the seated position at the end of the test. This value is roughly equivalent to the traditional TUG duration, with the difference that the identification of the onset, duration, and end of the test is based on the device’s algorithm rather than the examiner’s use of a stopwatch.-Walk duration including the half-turn [s] refers to the time interval between the algorithm-detected end of the sit-to-stand phase, when the subject begins to walk, and the return toward the chair, including the half-turn. The endpoint is defined as the start of the turn in which the subject initiates sitting.-Root Mean Square (RMS) of vertical acceleration during sit-to-walk [m/s^2^] is the RMS of the acceleration signal in the vertical direction during the sit-to-walk phase. It provides a measure of the overall amplitude of acceleration over time and reflects how dynamic the lower-limb effort is during sit-to-walk [24].-Range of the vertical angular velocity during sit-to-walk [°/s] is the difference between the maximum and minimum angular velocity around the vertical axis measured during the sit-to-walk phase. It quantifies how much the pelvis rotates around vertical axis.-RMS of the vertical angular velocity during sit-to-walk [°/s] is the RMS of the angular velocity around the vertical axis measured during the sit-to-walk phase. It reflects the overall magnitude or intensity of rotational movement of the pelvis.-Vertical push-off power during sit-to-walk [Nm] is an indirect, inertia-based estimate of the power generated during the sit-to-walk phase. It is derived from vertical acceleration data and is intended to reflect the ability of the lower limbs to generate muscular power during the postural transition from sitting to standing [17].

Concentric isokinetic strength data for the quadriceps and hamstrings of both the affected and healthy sides included torque values normalized to body mass [Nm/kg].

#### Bias

To minimize selection bias, the study was systematically proposed to all patients meeting eligibility criteria during the recruitment period. Demographic and clinical data were verified by a study investigator to ensure accuracy and completeness. To reduce performance bias, all evaluations were performed by trained personnel using a standardized procedure. A detailed testing protocol was available to all investigators. All functional and strength assessments were conducted on the same day, in a fixed order, with rest intervals provided between tests. The difficulty in using inertial units currently lies in signal processing and analysis, with the possible presence of artifacts, particularly during complex movements. The markers used for signal analysis vary widely across studies [28]. Furthermore, sensor positioning is a key element in the measurement process, since it influences the signal. Many possible positions exist, and several inertial units may be attached to an individual. Most studies have used a single inertial unit attached to the sacrum, which appears to be the optimum position [29]. The results highlighted in our work are based on statistical correlations with clinical hypotheses, but associations cannot be considered causal.

### 2.7. Sample Size

The sample size calculation was based on a Type I error (alpha) of 5% and a Type II error (beta) of 10%. With a Type I error (α) set at 5%, a statistical power of 90%, and the objective to detect a correlation coefficient of at least 0.5, the required minimum sample size was 38 participants.

### 2.8. Statistical Methods

Data are described by frequency and percentage for categorical data and by mean and standard deviation for continuous data (or median and interquartile range when the distribution was not normal). Relationships between continuous data were assessed using Spearman’s correlation coefficient. Correlation analysis is performed separately for the two distinct conditions due to the different clinical and biomechanical consequences of knee and hip osteoarthritis. Correlation coefficients (r) were interpreted as follows: values between 0.10 and 0.39 as weak, 0.40–0.69 as moderate, and >0.70 as strong correlations. All tests were two-sided and a *p*-value < 0.05 was considered statistically significant. Statistics were performed using Stata v16 (StataCorp, College Station, TX, USA).

## 3. Results

### 3.1. Participants

Fifty-seven patients were included in this study between 26 August 2020 and 3 November 2021. The flow chart is presented in Figure 2.

In total, 57 participants were included in the study, comprising 31 women (54.5%). The mean age was 65.6 ± 9.5 years. Among them, 36 participants had KOA and 21 had HOA. Clinical characteristics of each group, including KOOS, HOOS, and WOMAC scores, are detailed in Table 1.

### 3.2. Main Results

Fifty-seven participants performed the iTUG evaluation and isokinetic strength measurements. The mean (SD) traditional TUG duration for participants with KOA was 11.6 (3.0) s and, for those with HOA, it was 11.6 (4.0) s. Correlation coefficients with 95% confidence intervals for muscle strength normalized to body mass and iTUG parameters are presented in Table 2. The results reveal distinct correlation patterns between KOA and HOA. Specifically, quadriceps and hamstring muscle strength showed stronger correlations with iTUG parameters in patients with HOA compared with those with KOA.

#### 3.2.1. Patients with Knee Osteoarthritis

In patients with KOA, quadriceps muscle strength shows moderate, significant correlations with vertical push-off power during sit-to-walk, walk duration including the half-turn, and total iTUG duration for both the affected and the asymptomatic sides (Table 2). Similarly, hamstring muscle strength shows moderate, significant correlations with vertical push-off power during sit-to-walk for the affected and the asymptomatic sides (Table 2). No other significant correlations were found in patients with KOA.

#### 3.2.2. Patients with Hip Osteoarthritis

In patients with HOA, quadriceps muscle strength on the affected side was strongly correlated with vertical push-off power during sit-to-walk, walk duration including the half-turn, and total iTUG duration (Table 3). For quadriceps strength on the asymptomatic side, and bilateral hamstring strength, correlations with the same iTUG parameters were moderate to strong (Table 3). The Root Mean Square of the vertical acceleration during sit-to-walk and the Root Mean Square of the vertical angular velocity during sit-to-walk also showed significant moderate correlations with quadriceps strength on the affected side (Table 3). No other significant correlations were found in patients with HOA.

An example scatter plot that illustrates the Spearman’s correlation between total iTUG duration and isokinetic quadriceps strength on the affected side is presented in Appendix A. A negative relationship is observed in both groups, with the steeper correlation trend line for individuals with HOA suggesting a stronger correlation than for individuals with KOA.

In addition, a strong Spearman’s correlation coefficient was found between the iTUG total duration and the traditional stopwatch TUG (r = 0.91, *p* < 0.01).

No adverse events occurred during the study.

## 4. Discussion

The main finding is that the iTUG test can be used to estimate quadriceps and hamstring muscle strength in patients with KOA and HOA. Based on the stronger and more consistent correlations between iTUG parameters and quadriceps muscle strength in HOA, compared with the weak to moderate correlations observed in KOA, the iTUG appears to have greater utility for detecting muscle weakness in HOA. Contrary to our hypothesis, the Root Mean Square of the vertical acceleration during sit-to-walk was not the parameter most strongly related to quadriceps strength. It showed only a moderate correlation on the affected side in HOA and no significant correlations in KOA. The lack of significant correlations for most acceleration and angular velocity variables suggests that the kinematics of the sit-to-walk transition are only marginally affected by deficits in quadriceps or hamstring strength. On the other hand, vertical push-off power during sit-to-walk, walk duration including the half-turn, and total iTUG duration were the parameters most strongly related to quadriceps and hamstring strength, especially in patients with HOA. These findings, however, must be viewed with caution given the exploratory nature of the study and the fact that the target sample size was not reached.

Previous studies have shown that both HOA and KOA induce muscle weakness and functional impairments that, clinically, are more similar than different [3,30,31]. In our study, iTUG parameters reflected better quadriceps weakness in HOA. Different gait kinematic patterns may provide insight into these results, although joint kinematics were not directly assessed in this study. Specifically, greater knee flexion and reduced hip extension in patients with HOA compared with those with KOA could potentially influence iTUG parameters [32]. In patients with HOA, reduced hip flexion during chair rising may also play a role [33]. Joint kinematics were not assessed in our study, but this should be addressed in future research to better understand the factors that determine iTUG parameters.

Vertical push-off power during sit-to-walk emerged as the parameter most strongly associated with quadriceps and hamstring muscle strength, particularly in HOA. This is consistent with earlier findings showing reduced lower limb muscles power and its association with functional performance in individuals with HOA or KOA [34,35]. However, total iTUG duration showed a similarly strong association in HOA and moderate association in KOA. Moreover, the strong correlation we found between the traditional stopwatch TUG and the total iTUG duration supports the idea that even the traditional TUG reflects quadriceps and hamstring muscle strength. The finding that longer total TUG duration appears to reflect overall weakness of the quadriceps and hamstring muscles aligns with a study showing that poor TUG performance could predict sarcopenia in older adults [36].

From a clinical perspective, since total iTUG duration is strongly correlated with the traditional TUG, both may represent practical and cost-effective alternatives to isokinetic testing for estimating muscle weakness and monitoring rehabilitation progress. Using iTUG parameters could provide more detailed insights into functional performance. It could provide information on potential impairment in specific phases of the test that may require improvement, while the traditional TUG test remains a simple and clinically relevant tool reflecting overall functional performance. Both could help guide rehabilitation by identifying individuals who may benefit from strengthening exercises, especially in clinical situations where isokinetic evaluation is not available.

### Limitations

As previously noted, the primary limitation of this study is the small sample size, particularly within the HOA group. This restricts the generalizability of our findings and limits statistical power. Consequently, while the strong associations observed in HOA are promising, they require confirmation in larger cohorts to ensure generalizability. In addition, including participants with either KOA or HOA at different stages of disease progression may have increased variability within the sample and impacted the correlation analysis. However, this reflects the variation typically observed in patients attending physical and rehabilitation medicine departments.

Although the reliability of the iTUG was not evaluated, previous systematic reviews have shown that the iTUG is a reliable and valid assessment tool [29].

Beyond muscle strength, several confounding factors may also influence iTUG performance. Pain can modify motor strategies by limiting movement amplitude or velocity, independently of actual strength levels. A high BMI can affect iTUG performance not only by increasing the mechanical load on the lower limbs but also by potentially altering trunk and pelvic kinematics or increasing soft-tissue oscillations that may contaminate the IMU signal. Additionally, the use of walking aids and the presence of balance deficits or sensor placement may further influence iTUG measurements. These factors should be considered in future studies and in the interpretation of iTUG results.

## 5. Conclusions

The iTUG test seems to effectively capture functional consequences of lower-limb muscle weakness, with vertical push-off power and temporal parameters showing the strongest and most consistent associations with quadriceps and hamstring strength. These relationships were more pronounced in HOA than those in KOA. The iTUG test could be used to provide more detailed insight into the consequences of quadriceps and hamstring muscle weakness, whereas the traditional TUG test remains a simple and pertinent clinical tool, as both are related to quadriceps and hamstring strength. However, the small sample size is a major limitation of our study. Therefore, larger and more robust studies are needed to confirm our results. Moreover, when using the iTUG test, the presented confounding factors should be taken into consideration.

## Figures and Tables

**Figure 1 sensors-26-00142-f001:**
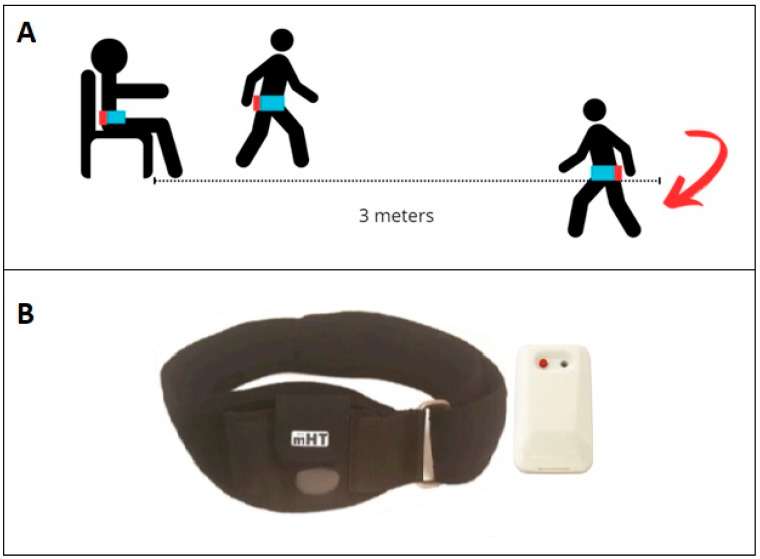
Setup for the iTUG test. (**A**) Schematic representation of the iTUG with mTUG^®^ sensor (mHealth Technologies, Bologna, Italy) placement (shown in red) secured by a belt (shown in blue); (**B**) belt and sensor components of the mTUG^®^ system. The arrow represents a U-turn.

**Figure 2 sensors-26-00142-f002:**
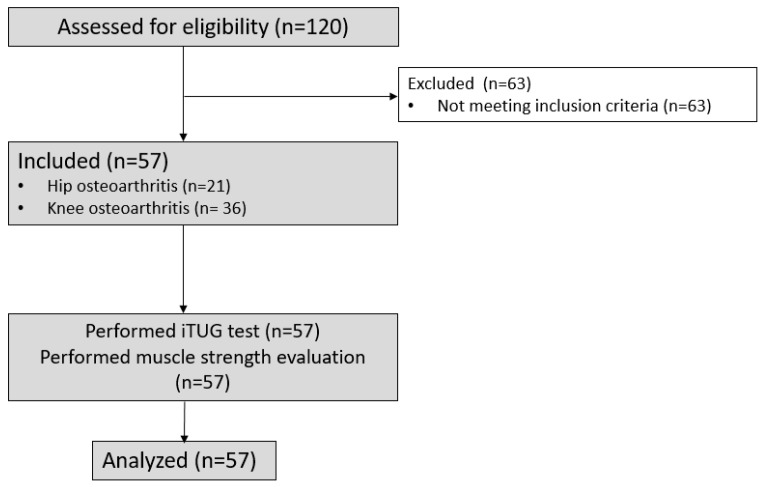
Flow chart.

**Table 1 sensors-26-00142-t001:** Participant characteristics (*n* = 57).

Variables	Knee Osteoarthritis (*n* = 36)	Hip Osteoarthritis (*n* = 21)
Sex: Female/male, n (%)	20 (55.6%)/16 (44.4%)	11 (52.4%)/10 (47.6%)
Age (years), mean (SD)	66.1 (7.8)	64.6 (12.1)
Height (cm), mean (SD)	165.4 (8.4)	164.4 (9.1)
Weight (kg), mean (SD)	87.4 (20.3)	81.0 (16.9)
BMI (kg/m^2^), mean (SD)	31.9 (6.7)	29.9 (5.4)
Type of residence: Flat/house, *n* (%)	8 (22.2%)/28 (77.8%)	5 (23.8%)/16 (76.2%)
Pathology, *n* (%)		
Osteoarthritis pre-chirurgical stage	31 (54.4%)	18 (31.6%)
Osteoarthritis medical stage	5 (8.8%)	3 (5.2%)
Timed Up and Go test (seconds), mean (SD)	11.6 (3.0)	11.6 (4.0)
KOOS, mean (SD)		
Symptoms	46.0 (15.5)	
Pain	47.8 (15.6)	
Function	49.8 (15.2)	
Activities	16.3 (18.6)	
Quality of life	26.4 (19.2)	
WOMAC score, mean (SD)		
Pain	49.2 (16.8)	
Stiffness	39.1 (19.5)	
Function	47.7 (16.5)	
HOOS, mean (SD)		
Symptoms	41.4 (19.7)	
Pain	41.9 (15.0)	
Daily activities	44.2 (18.4)	
Sporting activities	25.0 (21.5)	
Quality of life	23.2 (18.7)	

Data are mean (SD) or number and percentage. BMI: Body Mass Index; HOOS: Hip disability and Osteoarthritis Outcome Score; KOOS: Knee injury and Osteoarthritis Outcome Score; SD: standard deviation; WOMAC: Western Ontario and McMaster Universities Arthritis Index.

**Table 2 sensors-26-00142-t002:** Correlation analysis heatmap between iTUG parameters and isokinetic concentric quadriceps and hamstring muscle strength normalized to body mass at 60°/s in individuals with knee osteoarthritis (*n* = 36).

iTUG Parameters	Side	Muscle Strength	Correlation Coefficient
Root Mean Square of the Vertical Acceleration during Sit-to-Walk [m/s^2^]	Asymptomatic	Quadriceps	−0.124 [−0.435; 0.214]*p* = 0.472
Hamstring	−0.195 [−0.492; 0.143]*p* = 0.254
Affected	Quadriceps	−0.044 [−0.367; 0.289]*p* = 0.800
Hamstring	−0.129 [−0.439; 0.208]*p* = 0.452
Range of the Vertical Angular Velocity during Sit-to-Walk [°/s]	Asymptomatic	Quadriceps	0.028 [−0.303; 0.353]*p* = 0.870
Hamstring	0.108 [−0.228; 0.422]*p* = 0.529
Affected	Quadriceps	0.016 [−0.314; 0.342]*p* = 0.928
Hamstring	−0.095 [−0.411; 0.241]*p* = 0.581
Root Mean Square of the Vertical Angular Velocity during Sit-to-Walk [°/s]	Asymptomatic	Quadriceps	0.063 [−0.271; 0.384]*p* = 0.713
Hamstring	0.181 [−0.157; 0.481]*p* = 0.291
Affected	Quadriceps	0.128 [−0.21; 0.438]*p* = 0.458
Hamstring	0.042 [−0.29; 0.366]*p* = 0.807
Vertical Push-Off Power during Sit-to-Walk [Nm]	Asymptomatic	Quadriceps	−0.516 [−0.722; −0.225]***p* = 0.001**
Hamstring	−0.476 [−0.696; −0.175]***p* = 0.003**
Affected	Quadriceps	−0.456 [−0.682; −0.150]***p* = 0.005**
Hamstring	−0.396 [−0.641; −0.077]***p* = 0.017**
Walk Duration including the half-turn [s]	Asymptomatic	Quadriceps	−0.398 [−0.642; −0.079]***p* = 0.016**
Hamstring	−0.279 [−0.557; 0.054]*p* = 0.099
Affected	Quadriceps	−0.404 [−0.647; −0.088]***p* = 0.014**
Hamstring	−0.215 [−0.507; 0.123]*p* = 0.209
Total Duration [s]	Asymptomatic	Quadriceps	−0.409 [−0.65; −0.093]***p* = 0.013**
Hamstring	−0.310 [−0.579; 0.021]*p* = 0.066
Affected	Quadriceps	−0.410 [−0.651; −0.094]***p* = 0.013**
Hamstring	−0.251 [−0.536; 0.084]*p* = 0.139
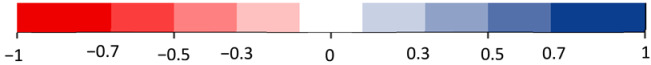

Spearman’s correlation coefficients include 95% confidence intervals and *p*-values. Bold values indicate significant results (*p* < 0.05). Red shades (from light pink to dark red) represent negative correlations, whose intensity increases as the color darkens (toward −1). Blue shades (from light to dark blue) indicate positive correlations, which become stronger as the color deepens (toward +1). White or neutral tones correspond to weak or no correlations (around 0).

**Table 3 sensors-26-00142-t003:** Correlation analysis heatmap between iTUG parameters and isokinetic concentric quadriceps and hamstring muscle strength normalized to body mass at 60°/s in individuals with hip osteoarthritis (*n* = 21).

iTUG Parameters	Side	Muscle Strength	Correlation Coefficient
Root Mean Square of the Vertical Acceleration during Sit-to-Walk [m/s^2^]	Asymptomatic	Quadriceps	−0.421 [−0.750; 0.075]*p* = 0.092
Hamstring	−0.230 [−0.640; 0.282]*p* = 0.374
Affected	Quadriceps	−0.537 [−0.809; −0.076]***p* = 0.026**
Hamstring	−0.253 [−0.654; 0.259]*p* = 0.327
Range of the Vertical Angular Velocity during Sit-to-Walk [°/s]	Asymptomatic	Quadriceps	0.235 [−0.277; 0.643]*p* = 0.363
Hamstring	0.311 [−0.199; 0.689]*p* = 0.223
Affected	Quadriceps	0.453 [−0.035; 0.767]*p* = 0.068
Hamstring	0.250 [−0.262; 0.652]*p* = 0.333
Root Mean Square of the Vertical Angular Velocity during Sit-to-Walk [°/s]	Asymptomatic	Quadriceps	0.328 [−0.181; 0.699]*p* = 0.198
Hamstring	0.360 [−0.146; 0.717]*p* = 0.155
Affected	Quadriceps	0.593 [0.157; 0.836]***p* = 0.012**
Hamstring	0.309 [−0.202; 0.687]*p* = 0.228
Vertical Push-Off Power during Sit-to-Walk [Nm]	Asymptomatic	Quadriceps	−0.725 [−0.894; −0.376]***p* = 0.001**
Hamstring	−0.623 [−0.849; −0.202]***p* = 0.008**
Affected	Quadriceps	−0.843 [−0.942; −0.609]***p* < 0.001**
Hamstring	−0.711 [−0.888; −0.350]***p* = 0.001**
Walk Duration including the half-turn [s]	Asymptomatic	Quadriceps	−0.641 [−0.857; −0.232]***p* = 0.006**
Hamstring	−0.592 [−0.835; −0.155]***p* = 0.012**
Affected	Quadriceps	−0.718 [−0.891; −0.363]***p* = 0.001**
Hamstring	−0.718 [−0.891; −0.363]***p* = 0.001**
Total Duration [s]	Asymptomatic	Quadriceps	−0.701 [−0.884; −0.332]***p* = 0.002**
Hamstring	−0.564 [−0.822; −0.114]***p* = 0.018**
Affected	Quadriceps	−0.819 [−0.932; −0.557]***p* < 0.001**
Hamstring	−0.684 [−0.876; −0.303]***p* = 0.002**
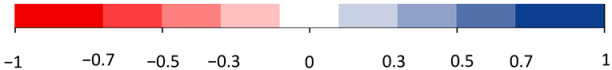

Spearman’s correlation coefficients include 95% confidence intervals and *p*-values. Bold values indicate significant results (*p* < 0.05). Red shades (from light pink to dark red) represent negative correlations, whose intensity increases as the color darkens (toward −1). Blue shades (from light to dark blue) indicate positive correlations, which become stronger as the color deepens (toward +1). White or neutral tones correspond to weak or no correlations (around 0).

## Data Availability

The datasets generated and analyzed during the current study are not publicly available due to ethical and legal restrictions related to participant confidentiality and the terms of approval by the ethics committee, but are available from the corresponding author upon reasonable request.

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
