# Peer review of "Sensors2026, 26(1), 142;https://doi.org/10.3390/s26010142"

_sensors, 2025, doi:10.3390/s26010142_

Round 1

Reviewer 1 Report

Comments and Suggestions for Authors

The manuscript addresses a clinically important topic related to the potential use of the instrumented Timed Up and Go (iTUG) test as an indirect indicator of quadriceps and hamstring muscle strength in individuals with hip and knee osteoarthritis. The study aims to explore to what extent parameters obtained from inertial sensors can serve as substitutes for isokinetic measurements, which represents a practical and potentially very useful approach in clinical settings.

Methodologically, the study is well designed and clearly described. It is a single-center cross-sectional study in which all tests were performed under standardized conditions, which contributes to the reliability of the results. It is also commendable that all measurements were carried out on the same day and by the same investigators, thereby minimizing variability. The combined use of the iTUG system and an isokinetic dynamometer allowed for a meaningful comparison between functional and biomechanical parameters, which is certainly one of the strengths of this work.

The results indicate weak to moderate negative correlations between iTUG parameters and isokinetic quadriceps and hamstring strength. The strongest associations were observed for total test duration and walking duration, an interesting finding suggesting that even the conventional “stopwatch-based” TUG test may provide valuable insight into lower-limb muscle strength. This finding has practical relevance for rehabilitation settings where isokinetic assessment is not available.

Although the study is methodologically solid and the results are clearly presented, several aspects could be further improved and clarified:

  • Although the procedure is described in detail, it would be much clearer for readers if at least one schematic illustration were added to show the sensor placement and test path. A study based on measurement technology deserves a visual element to support its explanation.
  • The manuscript lists the analyzed parameters (RMS of vertical acceleration, angular velocity, etc.), but it does not explain how the data were filtered, how the movement phases were segmented, or which measurement units were used. Providing these details would strengthen the reproducibility and credibility of the findings.
  • Including at least one scatter plot showing the relationship between iTUG duration and quadriceps strength would visually reinforce the statistical findings and make the results more convincing.
  • While the authors adequately summarize their main findings, the discussion would benefit from a broader reflection on the clinical implications of using iTUG as a practical alternative to isokinetic testing. It would also be useful to briefly address the potential limitations of this test when applied to patients with more severe functional impairments.
  • The sample size is relatively small and includes two clinically distinct groups (patients with hip and knee osteoarthritis), which may affect the homogeneity of the results. Factors such as pain level, physical activity, or the use of walking aids were not considered, although they could influence performance and he observed correlations.

In summary, this is a well-conceived paper with relevant clinical potential. By adding a few visual and methodological details, along with a slightly deeper discussion of clinical applicability, the manuscript would gain in clarity and overall impact.

Author Response

Reviewer n°1

The manuscript addresses a clinically important topic related to the potential use of the instrumented Timed Up and Go (iTUG) test as an indirect indicator of quadriceps and hamstring muscle strength in individuals with hip and knee osteoarthritis. The study aims to explore to what extent parameters obtained from inertial sensors can serve as substitutes for isokinetic measurements, which represents a practical and potentially very useful approach in clinical settings.

Methodologically, the study is well designed and clearly described. It is a single-center cross-sectional study in which all tests were performed under standardized conditions, which contributes to the reliability of the results. It is also commendable that all measurements were carried out on the same day and by the same investigators, thereby minimizing variability. The combined use of the iTUG system and an isokinetic dynamometer allowed for a meaningful comparison between functional and biomechanical parameters, which is certainly one of the strengths of this work.

The results indicate weak to moderate negative correlations between iTUG parameters and isokinetic quadriceps and hamstring strength. The strongest associations were observed for total test duration and walking duration, an interesting finding suggesting that even the conventional “stopwatch-based” TUG test may provide valuable insight into lower-limb muscle strength. This finding has practical relevance for rehabilitation settings where isokinetic assessment is not available.

Although the study is methodologically solid and the results are clearly presented, several aspects could be further improved and clarified:

  • Although the procedure is described in detail, it would be much clearer for readers if at least one schematic illustration were added to show the sensor placement and test path. A study based on measurement technology deserves a visual element to support its explanation.

We appreciate your comment. We concur with your suggestion and have consequently added a schematic illustration of the iTUG setup and the sensor with a belt fixation in Figure 1A and 1B. The following modification were introduced:

“The test and devide are shown schematically in Figure 1.”

“Figure 1. Setup for the iTUG test. (A) Schematic representation of the iTUG with mTUG® sensor placement (shown in red) secured by a belt (shown in blue); (B) Belt and sensor components of the mTUG® system.  “

  • The manuscript lists the analyzed parameters (RMS of vertical acceleration, angular velocity, etc.), but it does not explain how the data were filtered, how the movement phases were segmented, or which measurement units were used. Providing these details would strengthen the reproducibility and credibility of the findings.

The explanation on how movement phases were segmented are presented in the Methods (2.5.2). We added the succinct explanation with the references concerning determination of the specific technical parameters : “During the iTUG, the inertial measurement unit captures the duration of each TUG phase automatically (sit-to-walk, walking, half-turning, and stand-to-sit), as well as the corresponding 3D velocities and accelerations. The processes of filtering, segmenting the movement into phases, and defining the measurement parameters were based on previous studies.22,23

  • Including at least one scatter plot showing the relationship between iTUG duration and quadriceps strength would visually reinforce the statistical findings and make the results more convincing.

Thank you for this sugesstion. A scatter plot illustrating the relationship between total iTUG duration and isokinetic quadriceps strength on the affected side has been added to the revised manuscript (Supplementary File 1). The corresponding modification was added to the Results section:

“An example scatter plot that illustrates the Spearman correlation between total iTUG duration and isokinetic quadriceps strength on the affected side is presented in Supplementary File 1.”

  • While the authors adequately summarize their main findings, the discussion would benefit from a broader reflection on the clinical implications of using iTUG as a practical alternative to isokinetic testing. It would also be useful to briefly address the potential limitations of this test when applied to patients with more severe functional impairments.

We thank the reviewer for this valuable comment. We have revised the discussion to better highlight the clinical implications of using iTUG as a practical and accessible alternative to isokinetic testing. We have also added a short statement discussing the potential limitations of applying the iTUG in patients with more severe functional impairments. These modifications are now included in the revised Discussion section:

“From a clinical perspective, since total iTUG duration is strongly correlated with the traditional TUG, both may represent practical and cost-effective alternatives to isokinetic testing for estimating muscle weakness and monitoring rehabilitation progress. Using iTUG parameters could provide more detailed insights into functional performance, while the traditional TUG test remains a simple and clinically pertinent tool.”

  • The sample size is relatively small and includes two clinically distinct groups (patients with hip and knee osteoarthritis), which may affect the homogeneity of the results. Factors such as pain level, physical activity, or the use of walking aids were not considered, although they could influence performance and he observed correlations.

Thank you for this very pertinent remark. We agree that the sample size is relatively small and that hip and knee osteoarthritis represent two clinically distinct populations. Initially, our objective was to recruit 38 participants per group to allow adequately powered subgroup analyses. Due to feasibility constraints, we included 57 participants (36 with knee osteoarthritis and 21 with hip osteoarthritis). In light of the reviewer’s comment, and given that movement strategies and biomechanical demands differ between hip and knee OA, we performed separate correlation analyses for both groups and now report these results in the manuscript. These subgroup analyses indeed highlighted distinct clinical profiles: correlations were generally weak to moderate and inconsistent in knee osteoarthritis, whereas in hip osteoarthritis, associations between iTUG parameters and muscle strength were more robust and systematic. Consequently, we have considerably revised the manuscript according to this analysis. The modifications are listed below.

Methods, Statistical methods:

“Correlation analysis is performed separately for the two distinct conditions due to the different clinical and biomechanical consequences of knee and hip osteoarthritis.”

Results:

“Fifty-seven participants performed the iTUG evaluation and isokinetic strength measurements. The mean (SD) iTUG duration for all participants with KOA was 11.6 (3.0) seconds and for those with HOA was 11.6 (4.0) seconds.”

“3.2.1.   Patients with knee osteoarthritis

In patients with KOA, quadriceps muscle strength shows moderate, significant correlations with Vertical Push-Off Power during Sit-to-Walk, Walk Duration including the half-turn, and Total iTUG Duration (r = –0.398 to –0.516, p < 0.05) for both the affected and the asymptomatic sides. Similarly, hamstring muscle strength shows moderate, significant correlations with Vertical Push-Off Power during Sit-to-Walk for the affected side (r = –0.396, p = 0.017) and the asymptomatic side (r = –0.476, p = 0.003). No other significant correlations were found in patients with KOA. (Table 2.)

3.2.2.     Patients with hip osteoarthritis

In patients with HOA, quadriceps muscle strength on the affected side was strongly correlated with Vertical Push-Off Power during Sit-to-Walk, Walk Duration including the half-turn, and Total iTUG Duration (r = –0.718 to –0.843, p < 0.01). For quadriceps strength on the asymptomatic side, as well as for bilateral hamstring strength, correlations with Vertical Push-Off Power during Sit-to-Walk, Walk Duration including the half-turn, and Total iTUG Duration were all significant and moderate to strong (r = –0.592 to –0.725, p < 0.05). The Root Mean Square of the Vertical Acceleration during Sit-to-Walk and the Root Mean Square of the Vertical Angular Velocity during Sit-to-Walk also showed significant moderate correlations with quadriceps strength on the affected side (r = –0.537, p = 0.026 and r = 0.593, p = 0.012, respectively). No other significant correlations were found in patients with HOA. (Table. 2)”

The entire discussion, limitations and discussion have been modified accordingly.

In summary, this is a well-conceived paper with relevant clinical potential. By adding a few visual and methodological details, along with a slightly deeper discussion of clinical applicability, the manuscript would gain in clarity and overall impact.

Reviewer 2 Report

Comments and Suggestions for Authors

1.
"IMUs appear to be a wise choice for assessing
functional mobility because of their widespread availability and ease of use, even for un-
trained investigators."

It is true that IMUs are much simpler to use, but saying that untrained investigators can use them ignores the nuances that IMU placement and utilisation has. Perhaps the authors meant clinical experts that do not necessarily have a strong technical background (e.g. physiotherapists, nurses etc.) can use them without prior training

2.3
The number of participants and their mean age +/- St.D. should be mentioned at section 2.3 as well.

2.5
A picture of the mTug, as wel as relevant information regarding the model/year/company etc. is needed in section 2.5 when the name of the instrument is mentioned for the first time, not later in the text. Alternatively, the introductory paragraph of section 2.5 could refrain from stating the name of the software, similarly to how the ergometer's model is not mentioned there, and add the picture to 2.5.2.

Seoul is the capital of the Republic of Korea (colloquially South Korea), it is not in Japan. Samsung Group is a Korean company. 

Is the the smartphone attached using a belt or using tape?

2.6
RMS, in this case at least, is a measurement of how far the measured data stray from an estimation (e.g. the expected average or a different measurement). It has nothing to do with signal intensity. This also leads to the question with what each signal is compared? (average, affected/non-affected side etc.).

2.8
How was this number (38) was found? I assume some kind of power analysis was done, but it is not explained. 

3.2
Table 2
Are those  Pearson's correlation coefficients? In which cases are they Spearman's?
Also the red to blue scale could be improved by stating that it represents negative to positive correlation.
The addition of ** doesn't contribute anything since the p value is stated under the value.

"Given that most observed correlations were weaker than anticipated, the study may
have been underpowered to detect small to moderate associations, and findings should
be interpreted with caution" 
Is this perhaps a result from using people with different musculoskeletal issues (knee OA and hip OA) in the same sample?

4.
"Weiss et al. showed that iTUG parameters can better distinguish between fallers and
non-fallers among older adults. In contrast, our study found that the total duration of
the iTUG showed a stronger correlation with quadriceps strength than the individual
iTUG parameters. "

Your study did not mentions that history of falls was considered. Weiss considered idiopathic fallers, not people with OA. Based on the experiment that you presented, I don't see how your work is relevant to fall risk.

Similarly, "Moreover, as fall risk screening tools, information
about the need for muscle strengthening provided by the TUG or iTUG could help prevent
falls in older adults with knee or hip osteoarthritis" is not relevant. Also TUG is a measurement of performance, it may help in identification, but it can't hep on prevention.

5.
Your conclusion that TUG is equivalent to iTUG is not supported because there is no comparison between TUG/iTUG. It can be extrapolated since the isokinetic measurements have a moderate to weak correlation with the TUG results, but the reasoning is not explained in the text.  

In general, I think pooling together people with different types of OA really hurt the experiment. Splitting the populations might give a clearer picture.

Author Response

Reviewer n°2

1."IMUs appear to be a wise choice for assessing functional mobility because of their widespread availability and ease of use, even for un- trained investigators."

It is true that IMUs are much simpler to use, but saying that untrained investigators can use them ignores the nuances that IMU placement and utilisation has. Perhaps the authors meant clinical experts that do not necessarily have a strong technical background (e.g. physiotherapists, nurses etc.) can use them without prior training.

We thank the reviewer for this accurate interpretation. Yes, this is exactly what we meant, that IMUs can be easily used by clinical professionals such as physiotherapists or nurses, even if they do not have a strong technical or engineering background. We have revised the sentence to clarify this point in the Introduction:

“IMUs appear to be a practical choice for assessing functional mobility, due to their increasing availability, especially for clinical professionals such as physiotherapists and nurses, who may not have a strong technical background.”

2.3 The number of participants and their mean age +/- St.D. should be mentioned at section 2.3 as well.

Unless I am mistaken, I believe you are suggesting that we should provide details about the population in the Results section (part 3.1). All relevant participant characteristics including the number of participants and their mean age +/- St.D are now presented in Table 1 and in the text of the Results section.

2.5 A picture of the mTug, as wel as relevant information regarding the model/year/company etc. is needed in section 2.5 when the name of the instrument is mentioned for the first time, not later in the text. Alternatively, the introductory paragraph of section 2.5 could refrain from stating the name of the software, similarly to how the ergometer's model is not mentioned there, and add the picture to 2.5.2.

Seoul is the capital of the Republic of Korea (colloquially South Korea), it is not in Japan. Samsung Group is a Korean company. 

Is the the smartphone attached using a belt or using tape?

Thank you for your constructive comments.

As suggested, we have added a figure showing the inertial sensor and the adjustable belt used for attachment in Figure 1. The relevant model information was added in section 2.5 and the name of mTUG system was removed from Introduction. We also removed the specific smartphone model from Section 2.5, in order to be consistent with the description of the cycle ergometer.

In addition, we corrected the geographical reference, “Japan” has been replaced with “Korea” (Samsung Group, Seoul, Korea).

Finally, we clarified in Section 2.5.2 that the smartphone was held in the investigator’s hand during the test and was not attached to the participant in page 8. Also Figure 1A and 1B presents the clarification.

2.6 RMS, in this case at least, is a measurement of how far the measured data stray from an estimation (e.g. the expected average or a different measurement). It has nothing to do with signal intensity. This also leads to the question with what each signal is compared? (average, affected/non-affected side etc.).

Thank you for pointing out this issue. According to your pertinent remark, we have removed the sentence: “The RMS is a mathematical measure of signal intensity.” We have also specified in the Table affected/non-affected side. However, we did not perform comparisons of the average of each parameter between the groups, as this was not the objective of the study.

2.8 How was this number (38) was found? I assume some kind of power analysis was done, but it is not explained. 

We thank the reviewer for this comment. The sample size calculation was indeed based on a power analysis. As recommended, we have now detailed the parameters and reference used to determine the required number of participants. The calculation was performed according to Falissard (2005), with a type I error (α) set at 5%, a statistical power of 90%, and an expected correlation coefficient of at least 0.5, leading to a required minimum sample size of 38 participants.

This is now precise in Method, Sample size section: “With a type I error (α) set at 5%, a statistical power of 90%, and the objective to detect a correlation coefficient of at least 0.5, the required minimum sample size was 38 participants.”

3.2
Table 2

Are those  Pearson's correlation coefficients? In which cases are they Spearman's?
Also the red to blue scale could be improved by stating that it represents negative to positive correlation.
The addition of ** doesn't contribute anything since the p value is stated under the value.

Thank you for your comment. These are Spearman correlation coefficients presented in actual Table 2. As suggested by the reviewers, we removed the correlation analysis performed on the combined KOA+HOA group and now present the results separately for KOA and HOA. We have also removed the asterisks and added a legend:

 “The results are Spearman correlation coefficients with 95% confidence intervals and p-values. Values in bold indicate significant (p<0.05) results. The red shades (from light pink to dark red) represent negative correlations, whose intensity increases as the color darkens (toward -1). The blue shades (from light to dark blue) indicate positive correlations, which become stronger as the color deepens (toward +1). White or neutral tones correspond to weak or no correlations (around 0).”

Clarifications about the use of Spearman correlations were also included in the text.

Methods: “Relationships between continuous data were assessed using Spearman’s correlation coefficient”

Results: “An example scatter plot that illustrates the Spearman correlation between total iTUG duration and isokinetic quadriceps strength on the affected side is presented in Supplementary File 1.[…] In addition, a strong Spearman correlation coefficient was found between the iTUG total duration and the traditional stopwatch TUG (r = 0.91, p < 0.01).”

"Given that most observed correlations were weaker than anticipated, the study may
have been underpowered to detect small to moderate associations, and findings should
be interpreted with caution" 
Is this perhaps a result from using people with different musculoskeletal issues (knee OA and hip OA) in the same sample?

Thank you for this pertinent remark. Indeed, this issue was raised by all reviewers, and we greatly appreciate your insight. This concern significantly impacts our results, and we have revised the manuscript to present the correlation analyses separately for KOA and HOA. The modifications have been listed above.

  1. "Weiss et al. showed that iTUG parameters can better distinguish between fallers and
    non-fallers among older adults. In contrast, our study found that the total duration of
    the iTUG showed a stronger correlation with quadriceps strength than the individual
    iTUG parameters. "

Your study did not mentions that history of falls was considered. Weiss considered idiopathic fallers, not people with OA. Based on the experiment that you presented, I don't see how your work is relevant to fall risk.

Similarly, "Moreover, as fall risk screening tools, information
about the need for muscle strengthening provided by the TUG or iTUG could help prevent
falls in older adults with knee or hip osteoarthritis" is not relevant. Also TUG is a measurement of performance, it may help in identification, but it can't hep on prevention.

We thank the reviewer for this important clarification. The corresponding sentences have been removed.

5.Your conclusion that TUG is equivalent to iTUG is not supported because there is no comparison between TUG/iTUG. It can be extrapolated since the isokinetic measurements have a moderate to weak correlation with the TUG results, but the reasoning is not explained in the text.  

Thank you for pointing out this problem. We added in Method section 2.5.2 that a traditional stopwatch TUG measurement was also performed.

“A traditional stopwatch TUG measurement was performed simultaneously by the investigator.”

The results of traditional stopwatch TUG measurement are presented in Table 1. To address your pertinent remark, we also performed an additional correlation analysis between the iTUG total duration and the traditional stopwatch TUG. This supplementary analysis has been added to the Results section.

“In addition, a strong Spearman correlation coefficient was found between the iTUG total duration and the traditional stopwatch TUG (r = 0.91, p < 0.01).”

Finally, we adapted also corresponding sentences in Discussion section:

“From a clinical perspective, since total iTUG duration is strongly correlated with the traditional TUG, both may represent practical and cost-effective alternatives to isokinetic testing for estimating muscle weakness and monitoring rehabilitation progress. Using iTUG parameters could provide more detailed insights into functional performance, while the traditional TUG test remains a simple and clinically pertinent tool.”

In Conclusion:

“The iTUG test effectively captures functional consequences of lower-limb muscle weakness, with vertical push-off power and temporal parameters showing the strongest and most consistent associations with quadriceps and hamstring strength. These relationships were particularly robust in HOA, whereas associations in KOA were weaker and more heterogeneous. The iTUG test can be used to provide more detailed insight into the consequences of quadriceps and hamstring muscle weakness, whereas the traditional TUG test remains a simple and pertinent clinical tool, as both are related to quadriceps and hamstring strength. However, the small sample size is a major limitation of our study. Therefore, larger and more adequately powered studies are needed to verify our results. Moreover, when using the iTUG test, the presented confounding factors should be taken into consideration.”

In general, I think pooling together people with different types of OA really hurt the experiment. Splitting the populations might give a clearer picture.

Thank you for pointing out this important issue, which was raised by all reviewers. This concern significantly impacts our results, and we have removed the correlation analysis performed on the combined KOA+HOA group and now present the results separately for KOA and HOA. The modifications have been listed above.

Reviewer 3 Report

Comments and Suggestions for Authors

This manuscript presents an exploratory cross-sectional study examining the correlations between instrumented Timed Up and Go (iTUG) test parameters and isokinetic quadriceps/hamstring strength in individuals with hip or knee osteoarthritis (OA).

1.Provide a post-hoc power analysis or confidence intervals for the main correlation coefficients to clarify the robustness of findings.
2.Present subgroup analyses (knee vs. hip OA) for key outcomes or at least include a supplementary table to show whether correlations differ meaningfully between groups.
3.Elaborate on the biomechanical reasoning behind this choice and discuss potential confounding from sensor placement or trunk movement artifacts.
4.Reframe this conclusion to emphasize that iTUG offers quantitative insight while confirming that traditional TUG remains clinically useful.
5.Standardize all units and symbols throughout the text and figures for clarity and consistency.
6.The English is overall clear but would benefit from minor stylistic editing to improve fluency.
7.Explicitly discuss potential confounding effects of pain, BMI, or balance deficits on iTUG performance.
8.Specify whether both limbs were tested in all participants and whether the “affected” limb was defined clinically or radiographically.
9.Clarify if the same chair height was used for all subjects during iTUG testing.
10.The abstract is concise but could include numerical correlation values to provide more quantitative insight.

Author Response

Reviewver n°3

This manuscript presents an exploratory cross-sectional study examining the correlations between instrumented Timed Up and Go (iTUG) test parameters and isokinetic quadriceps/hamstring strength in individuals with hip or knee osteoarthritis (OA).

1.Provide a post-hoc power analysis or confidence intervals for the main correlation coefficients to clarify the robustness of findings.

Thank you for your meticulous reading of the manuscript and for this comment. Unfortunatelly, post-hoc power analysis is not of great help to assess robustness of the findings. Let consider a very strong correlation coefficient (like the r=0.91 for the iTUG / TUG correlation). The post-hoc power of this correlation is very strong. In contrast, a very low correlation coefficient will imply a low post-hoc power, even in the case of a very large sample size. This is because post-hoc power is more dependant of effect size than sample size. Thus, post-hoc power interpretation is not a valuable indicator of real power.

In order to interpret power after the completion of a study, it is more relevant to present confidence intervalls, as the more power leads to tighter intervall and vice versa. (doi: 10.3899/jrheum.211115.)

2.Present subgroup analyses (knee vs. hip OA) for key outcomes or at least include a supplementary table to show whether correlations differ meaningfully between groups.

Thank you for pointing out this important issue, which was raised by all reviewers. This concern significantly impacts our results, and we have revised the manuscript to present the correlation analyses separately for KOA and HOA. The modifications have been listed above.

3.Elaborate on the biomechanical reasoning behind this choice and discuss potential confounding from sensor placement or trunk movement artifacts.

We thank the reviewer for this pertinent suggestion. As explained above, we have revised this choice and made the corresponding modifications.

4.Reframe this conclusion to emphasize that iTUG offers quantitative insight while confirming that traditional TUG remains clinically useful.

Thank you for this suggestion. We have added a sentence to this effect in the Discussion.

“Using iTUG parameters could provide more detailed insights into functional performance, while the traditional TUG test remains a simple and clinically pertinent tool.”

And Conclusion:

“The iTUG test can be used to provide more detailed insight into the consequences of quadriceps and hamstring muscle weakness, whereas the traditional TUG test remains a simple and pertinent clinical tool, as both are related to quadriceps and hamstring strength.”

5.Standardize all units and symbols throughout the text and figures for clarity and consistency.

Thank you for pointing out this problem. This is now corrected.

6.The English is overall clear but would benefit from minor stylistic editing to improve fluency.

We hope that the provided modifications have enhanced the manuscript’s style and fluency.

7.Explicitly discuss potential confounding effects of pain, BMI, or balance deficits on iTUG performance.

We thank the reviewer for this valuable suggestion. We have now expanded the Discussion to explicitly address potential confounding effects of pain, body mass index (BMI), and balance deficits on iTUG performance.The revised text has been added to the Discussion section.

8.Specify whether both limbs were tested in all participants and whether the “affected” limb was defined clinically or radiographically.

We thank the reviewer for this relevant remark. Both limbs were tested in all participants. The affected limb was defined based on both clinical examination (pain, stiffness, and functional limitation) and radiographic confirmation of osteoarthritis severity. This clarification has been added in the Methods section part 2.5.1.

“Both limbs were tested in all participants. The affected limb was identified based on clinical findings (pain, stiffness, functional limitation) and confirmed by radiographic evidence of osteoarthritis.”

9.Clarify if the same chair height was used for all subjects during iTUG testing.

We added the clarification: in Methods section, part 2.5.2.:  “The same 45-cm chair with armrests was used for all participants.”

10.The abstract is concise but could include numerical correlation values to provide more quantitative insight.

Thank you for your comment. The abstract has been completely reworked following the modification of our article.

Round 2

Reviewer 2 Report

Comments and Suggestions for Authors

All comments have been addressed

Author Response

Reviewer 2

All comments have been addressed.

Reviewer 3 Report

Comments and Suggestions for Authors

This revised manuscript presents a well-structured cross-sectional study examining the relationship between instrumented Timed Up and Go (iTUG) parameters and isokinetic muscle strength in individuals with knee and hip osteoarthritis. 1.The title is clear but could be more concise. Consider: “Correlation between instrumented Timed Up and Go performance and lower-limb muscle strength in knee and hip osteoarthritis: a cross-sectional study.” 2.The abstract should explicitly state the study design in the first sentence. Specify that correlations were “weak-to-moderate” in KOA and “moderate-to-strong” in HOA for better clarity. The clinical implication regarding the traditional TUG is somewhat overstated. Consider phrasing: “The classic stopwatch-based TUG may offer comparable clinical insights to certain iTUG-derived metrics…” 3.The rationale for focusing on vertical acceleration during sit-to-walk is well explained, but a clearer link between biomechanical theory and clinical applicability would strengthen the section. Consider adding a sentence on the novelty of this study relative to existing iTUG literature in OA populations. 4.Section 2.5.1: Clarify whether the same protocol was used for hip and knee OA patients despite potential joint-specific limitations. Section 2.5.2: Mention whether practice trials were allowed before the iTUG test. Section 2.6: The description of iTUG parameters should include a brief explanation of how they were derived from sensor data. Section 2.8 (Sample size): It is noted that the target sample was not fully met, especially for HOA. This should be explicitly acknowledged as a limitation in the Discussion. 5.Table 2 is informative but visually dense. Consider splitting into two tables (KOA and HOA) for readability. In text (Section 3.2), avoid repeating all correlation coefficients; summarize key trends and refer to the table. The scatter plot in Supplementary File 1 should be mentioned in the main text with a brief interpretation. 6.The stronger correlations in HOA versus KOA are discussed, but the pathophysiological and biomechanical explanations remain speculative. Consider tempering conclusions given the exploratory nature and small subgroup samples. The paragraph on clinical implications (Page 19) should more clearly distinguish between the iTUG (detailed parameters) and traditional TUG (global duration) in terms of clinical utility. The limitation regarding sample size should be moved earlier in the Discussion and expanded upon, including its impact on correlation stability and generalizability. 7.The manuscript is generally well-written, but there are occasional awkward phrasings and minor grammatical errors. Consistent use of abbreviations after first mention. Some sentences are overly long; consider splitting for readability.

Author Response

Reviewer 3

This revised manuscript presents a well-structured cross-sectional study examining the relationship between instrumented Timed Up and Go (iTUG) parameters and isokinetic muscle strength in individuals with knee and hip osteoarthritis.

1.The title is clear but could be more concise. Consider: “Correlation between instrumented Timed Up and Go performance and lower-limb muscle strength in knee and hip osteoarthritis: a cross-sectional study.”

We thank the reviewer for this suggestion. We agree that the proposed formulation is more concise while preserving scientific clarity. We have therefore adopted the recommended title.

2.The abstract should explicitly state the study design in the first sentence. Specify that correlations were “weak-to-moderate” in KOA and “moderate-to-strong” in HOA for better clarity. The clinical implication regarding the traditional TUG is somewhat overstated. Consider phrasing: “The classic stopwatch-based TUG may offer comparable clinical insights to certain iTUG-derived metrics…”

Thank you for this valuable comment. We revised the abstract; however, we believe that the first two sentences should explain why the study was conducted. Immediately after, the third sentence presents the study design.

“This cross-sectional study investigated [...]”

According you suggestion, we specified that correlations were “weak-to-moderate” in KOA and “moderate-to-strong” in HOA.

“Correlations were weak-to-moderate in KOA (r = –0.398 to –0.516, p < 0.05) and moderate-to-strong in HOA (r= -0.537 to -0.843, p<0.05).”

The clinical implications regarding the traditional TUG have been toned down, and the study’s limitations have been clearly stated.

“Total iTUG duration strongly correlated with the traditional stopwatch TUG (r = 0.91, p < 0.01), suggesting that the classic stopwatch-based TUG may offer comparable clinical insights to certin iTUG-derived temporal metrics. The iTUG test reflects lower-limb muscle weakness, especially in HOA, with vertical push-off power and temporal metrics showing the strongest relationships. These results support the use of iTUG as accessible tool for estimating muscle weakness when isokinetic testing is unavailable. However, these findings should be interpreted with caution given the small sample size.”

3.The rationale for focusing on vertical acceleration during sit-to-walk is well explained, but a clearer link between biomechanical theory and clinical applicability would strengthen the section. Consider adding a sentence on the novelty of this study relative to existing iTUG literature in OA populations.

Thank you for this remark. We added a brief clarification in the Introduction regarding the link to clinical applicability and the novelty of this study relative to existing iTUG literature.

“Several studies have evaluated the use of inertial measurement units with various sensor placements in older adults as well as in individuals with KOA and HOA.13,15,16 However, no studies have examined the correlation between isokinetic quadriceps and hamstring muscle strength and the parameters derived from the mTUG® inertial measurement unit in individuals with KOA and HOA. Such an analysis could provide insight into the clinical relevance of using this device to detect muscle weakness.”

4.Section 2.5.1: Clarify whether the same protocol was used for hip and knee OA patients despite potential joint-specific limitations.

Thank you for these constructive remarks. We have clarified this point in section 2.5. as follows:

“The same testing procedure was applied to all participants, regardless of whether they had KOA or HOA.”

Section 2.5.2: Mention whether practice trials were allowed before the iTUG test.

In Section 2.5.2, we now indicate that participants did not perform a practice trial before the iTUG.

“The test was explained verbally by the investigator; however, no practice trial was performed.”

Section 2.6: The description of iTUG parameters should include a brief explanation of how they were derived from sensor data.

Thank you for this constructive comment. We agree that the description of iTUG-derived parameters required further clarification. Based on the specifications provided by the device manufacturer, we have now added a concise explanation on how each parameter is derived from the raw inertial sensor data. The revised text clarifications in section 2.6 are:

“Data collected from the iTUG evaluation were obtained from raw tri-axial accelerometer and gyroscope signals, which were processed through validated algorithms that automatically segment the iTUG into sit-to-walk, walking, turning, and walk-to-sit phases.28 These algorithms identify movement onset, transitions between phases, and movement termination, enabling the extraction of time- and amplitude-based metrics. Following iTUG parameters were recorded:

-           Total iTUG duration [s], which represents the time interval between the algorithm-detected onset of the sit-to-stand phase and the return to the seated position at the end of the test. This value is roughly equivalent to the traditional TUG duration, with the difference that the identification of the onset, duration, and end of the test is based on the device’s algorithm rather than the examiner’s use of a stopwatch.

 -          Walk duration including the half-turn [s] refers to the time interval between the algorithm-detected end of the sit-to-stand phase, when the subject begins to walk, and the return toward the chair, including the half-turn. The endpoint is defined as the start of the turn in which the subject initiates sitting.

-           Root mean square (RMS) of vertical acceleration during sit-to-walk [m/s²] is the RMS of the acceleration signal in the vertical direction during the sit-to-walk phase. It provides a measure of the overall amplitude of acceleration over time and reflects how dynamic the lower-limb effort is during sit-to-walk.28

-           Range of the vertical angular velocity during sit-to-walk [°/s] is the difference between the maximum and minimum angular velocity around the vertical axis measured during the sit-to-walk phase. It quantifies how much the pelvis rotates around vertical axis.

-           RMS of the vertical angular velocity during sit-to-walk [°/s] is the RMS of the angular velocity around the vertical axis measured during the sit-to-walk phase. It reflects the overall magnitude or intensity of rotational movement of the pelvis.

-           The Vertical push-off power during sit-to-walk [Nm] is an indirect, inertia-based estimate of the power generated during the sit-to-walk phase. It is derived from vertical acceleration data and is intended to reflect the ability of the lower limbs to generate muscular power during the postural transition from sitting to standing.29

Section 2.8 (Sample size): It is noted that the target sample was not fully met, especially for HOA. This should be explicitly acknowledged as a limitation in the Discussion.

In the Discussion (Limitations section), we added an explicit statement noting that the a priori sample size target was not reached, particularly for hip osteoarthritis.

“As previously noted, the primary limitation of this study is the small sample size, particularly within the HOA group. This restricts the generalizability of our findings and limit statistical power. Consequently, while the strong associations observed in HOA are promising, they require confirmation in larger cohorts to ensure generalizability.”

And also, mentioned this limitation earlier in the Discussion:

“These findings, however, must be viewed with caution given the exploratory nature of the study and the fact that the target sample size was not reached.”

5.Table 2 is informative but visually dense. Consider splitting into two tables (KOA and HOA) for readability.

Thank you for your comment. Indeed, this table is very dense. Following your comment, we have divided it into two: Table 2 for KOA and Table 3 for HOA.

In text (Section 3.2), avoid repeating all correlation coefficients; summarize key trends and refer to the table.

According to you remark, we have removed the correlation values from the text in Section 3.2 and now refer to the tables instead.

The scatter plot in Supplementary File 1 should be mentioned in the main text with a brief interpretation.

Thank you for your comment. We have added a brief explanation of the results presented in Supplementary File 1 in the text:

“A negative relationship is observed in both groups, with the steeper correlation trend line for individuals with HOA suggesting a stronger correlation than for individuals with KOA.”

6.The stronger correlations in HOA versus KOA are discussed, but the pathophysiological and biomechanical explanations remain speculative. Consider tempering conclusions given the exploratory nature and small subgroup samples.

We agree with the reviewer that our biomechanical explanations are hypothesis-driven and not directly supported by joint kinematic measurements. However, we believe that hypothetical explanations of the obtained results are a standard part of the Discussion section in research articles and are intended to point out what should be addressed in future studies. We have revised the Discussion section to temper our conclusions by using conditional language and by clearly stating that the small sample size remains an important limitation.

“Based on the stronger and more consistent correlations between iTUG parameters and quadriceps muscle strength in HOA, compared with the weak to moderate correlations observed in KOA, the iTUG appears to have greater utility for detecting muscle weakness in HOA. […] These findings, however, must be viewed with caution given the exploratory nature of the study and the fact that the target sample size was not reached. […] Different gait kinematic patterns may provide insight into these results, although joint kinematics were not directly assessed in this study. Specifically, greater knee flexion and reduced hip extension in patients with HOA compared with those with KOA could potentially influence iTUG parameters.34  In patients with HOA, reduced hip flexion during chair rising may also play a role.35 Joint kinematics were not assessed in our study, but this should be addressed in future research to better understand the factors that determine iTUG parameters.”

The Conclusion section has also been revised accordingly.

“The iTUG test seems to effectively capture the functional consequences of lower-limb muscle weakness, with vertical push-off power and temporal parameters showing the strongest and most consistent associations with quadriceps and hamstring strength. These relationships were more pronounced in HOA, than those in KOA. The iTUG test could be used to provide more detailed insight into the consequences of quadriceps and hamstring muscle weakness, […]”

Abstract:

“These results support the use of iTUG as accessible tool for estimating muscle weakness when isokinetic testing is unavailable. However, these findings should be interpreted with caution given the small sample size.”

Highlights:

“The iTUG test could provide a simple and accessible estimation of lower-limb muscle strength when isokinetic evaluation is not available.”

The paragraph on clinical implications (Page 19) should more clearly distinguish between the iTUG (detailed parameters) and traditional TUG (global duration) in terms of clinical utility.

Thank you for pointing out this issue. We have modified this section to provide a clearer distinction between the clinical implications of the iTUG and the traditional TUG:

“Using iTUG parameters could provide more detailed insights into functional performance. It could provide information on potential impairment in specific phases of the test that may require improvement, while the traditional TUG test remains a simple and clinically relevant tool reflecting overall functional performance.”

The limitation regarding sample size should be moved earlier in the Discussion and expanded upon, including its impact on correlation stability and generalizability.

According to your remark, we have added a concise statement earlier in the Discussion, second paragraph:

“These findings, however, must be viewed with caution given the exploratory nature of the study and the fact that the target sample size was not reached.”

and further developed it in the Limitations section of the Discussion:

“As previously noted, the primary limitation of this study is the small sample size, particularly within the HOA group. This restricts the generalizability of our findings and limit statistical power. Consequently, while the strong associations observed in HOA are promising, they require confirmation in larger cohorts to ensure generalizability.”

7.The manuscript is generally well-written, but there are occasional awkward phrasings and minor grammatical errors. Consistent use of abbreviations after first mention. Some sentences are overly long; consider splitting for readability.

Thank you for your comment, which encourages us to improve our manuscript. We have therefore revised the wording of certain sentences, reduced the length of sentences, and corrected the English.